# In Vitro Study of Butyric Acid Deodorization Potential by Indigenously Constructed Bacterial Consortia and Pure Cultures from Pit Latrine Fecal Sludge

**John Bright Joseph Njalam'mano \*, Evans Martin Nkhalambayausi Chirwa**  **and Refilwe Lesego Seabi**

Water Utilisation and Environmental Engineering Division, Department of Chemical Engineering, University of Pretoria, Private Bag X20, Hatfield, Pretoria 0028, South Africa; evans.chirwa@up.ac.za (E.M.N.C.); lseabi@gmail.com (R.L.S.)

\* Correspondence: chinamvuu@yahoo.co.uk; Tel.: +27-12-420-5894

**Abstract:** The present study aims at developing an efficient bacterial consortium to biodegrade butyric acid, one of the odor-causing compounds that contribute significantly to pit latrine malodors. Six bacterial strains isolated from pit latrine fecal sludge were selected for the study. Nineteen bacterial consortia of different combinations were artificially constructed. The individual bacterial strains and bacterial consortia were compared by culturing in mineral salt medium supplemented with 1000 mg/L butyric acid as a sole carbon and energy source at pH 7, 30 °C, and 110 rpm under aerobic growth conditions. A co-culture of *Serratia marcescens* and *Bacillus cereus* was an effective bacterial consortium compared to individual component bacterial strains and other bacterial consortia, in which 1000 mg/L butyric acid was completely degraded within 16 h of incubation. A temperature of 30 °C and pH 7 were found to be optimum for the maximum degradation for both *S. marcescens* and *B. cereus*. The inoculation sizes of 2.0 and 2.5 were optimal for the maximum degradation for *B. cereus* and *S. marcescens*, respectively. The study provides insights that will be of substantial help in the development of effective biological treatment technologies for pit latrine odor to change the pit latrine user community's and would be users' perception of pit latrines.

**Keywords:** *Bacillus cereus*; bacterial consortia; biodegradation; odor-causing compounds; sanitation; *Serratia marcescens*

## 1. Introduction

United Nations member states adopted the 2030 Agenda for Sustainable Development in September, 2015 to replace the expired Millennium Development Goals [1]. The Millennium Development Goal sanitation targets were not achieved and were replaced with a more ambitious Sustainable Development Goal (SDG) 6 sanitation target that aims to achieve access to adequate and equitable sanitation for all by 2030. Globally, there are 2.4 billion people that have no access to improved sanitation [1]. The worst affected people are living in the informal urban settlements and rural areas in developing countries [2]. A lack of access to sanitation facilities compels people to practice open defecation. Global estimates indicate that approximately 946 million people, without access to improved sanitation, habitually practice open defecation [1]. This practice facilitates the spread of water and sanitation related diseases, such as cholera, typhoid, hepatitis, polio, cryptosporidiosis, ascariasis schistosomiasis, and others [3] that precipitate morbidity and mortality, especially in developing countries [4].

The relative importance of sanitation in the interruption of pathogen transmission through the proper disposal of human excreta together with minimum levels of personal and domestic hygiene have been reported in literature. [5,6]. Additionally, an investment in sanitation is not only crucial for improved human health, but also offers a foundation for economic growth and social gains [7]. In developing countries, especially in the peri-urban and rural areas, pit latrines remain a predominant method of providing on-site sanitation facilities [8]. Even though pit latrines are the lowest rung on the sanitation ladder, they are utilized by an estimated 1.77 billion people worldwide [9]. The reality in the developing countries is that the number of pit latrines is anticipated to increase either as a transitional or permanent standalone solution or in combination with other improved human excreta disposal systems if the SDG 6 sanitation target is to be met. However, pit latrines are not without performance limitations. Malodors emanating from the pit are one of the performance limitations associated with pit latrines. Malodors are reported to be one of the most important determinants of the investment, adoption, and consistent use of pit latrines [10–12].

Butyric acid is one of the predominant odor-causing compounds detected in pit latrine emissions [13]. In its pure state, it has an extremely pungent, sweet, and rancid smell that makes it quite difficult to tolerate [14]. Butyric acid, a four-carbon volatile organic compound, is an intermediate product generated in the anaerobic bioconversion of complex organic matter to methane and carbon dioxide [14]. Some pit latrine fecal portions are anaerobic accumulation systems for stabilizing both human wastes, grey water, and household solid wastes [15,16]. This means that they can emit malodorous compounds, including butyric acid [17]. The mechanism for the generation of malodorous compounds, including butyric acid, are described by Mainville [18]. To promote the uptake and consistent use of pit latrines and, consequently, reduce the practice of open defecation in developing countries, it is a necessity to develop appropriate low-cost and intrinsically environmentally friendlier pit latrine deodorization techniques. These techniques may attenuate or eliminate the odor-causing compounds, such as butyric acid, to realize the benefits of sanitation.

Numerous pro-poor techniques and strategies have been devoted to the elimination of unpleasant smells in developing countries. These include the use of naturally scented substances, wood ash, saw dust, disinfectants, dry grass, husks, pesticides, oil, laundry and soapy water, detergents, and car-battery acids. Modified latrine designs, such as a ventilated improved pit (VIP) latrine, urine-diversion, and composting toilets and water seal latrines, etc., have also been used [10,19,20]. Although these techniques and strategies are available, they are associated with their own social, economic, institutional, and technological challenges, which have often made them a less desirable choice.

Bioremediation is the use of microorganisms to transform or mineralize hazardous organic materials into harmless or less hazardous compounds [21]. In this process, microorganisms obtain energy from the oxidation of primary substrates—i.e., carbon—which are converted into innocuous end products, such as carbon dioxide ($CO_2$), water ($H_2O$), inorganic salts, some volatile organic compounds, and microbial biomass, by assimilating part of the carbon into new cell material [22]. Microorganisms play a significant role in degrading pollutants in the environment [23]. Microorganisms capable of degrading malodorous compounds may be an attractive alternative to the existing odor control techniques and strategies currently used in low-income settings. Previous studies [24–26] have found that many bacterial strains can degrade butyric acid. However, despite the increased attention internationally in bioremediation, there is limited information on the biodegradation of butyric acid in the environment, particularly in pit latrine fecal sludge.

The importance of reducing malodors in pro-poor sanitation technologies, such as pit latrines, has not received much attention, despite their widespread use with respect to developing countries. Most of the literature related to odor reduction in wastewater focuses on high-tech sanitation technologies. Our previous work showed that *Achromobacter xylosoxidans*, *Bacillus subtilis*, *Lysinibacillus fusiformis*, *B. cereus*, *P.Pseudomonas aeruginosa*, *Bacillus methylotrophicus*, *S. marcescens*, *Achromobacter animicus*, and *Alcaligenes* sp. strain SY1 isolated from pit latrine fecal sludge samples

could grow using butyric acid as a sole source of carbon and energy [27]. However, bacteria in natural environments do not live in seclusion, but they dynamically interact with many other bacterial strains in complex multispecies communities [28]. Precisely for this reason, the present study seeks to address the following research questions: (1) do the constructed bacterial consortia enhance butyric acid degradation or not? (2) How do selected environmental factors of initial inoculation size, temperature, and pH affect the performance of individual strains of the most efficient bacterial consortium? The following objectives were derived from the research questions: to evaluate and compare the butyric acid degradation efficiencies of the bacterial consortia formulated using different combinations and their respective individual bacterial strains; to evaluate the effect of initial inoculation concentration, temperature, and pH on the growth and butyric acid degradation efficiencies of the constituent bacterial strains of the most efficient bacterial consortium.

## 2. Materials and Methods

### 2.1. Bacterial Cultures

The bacterial strains used in this work were isolated from pit latrine fecal sludge in Mpumalanga Province, South Africa (26°5′24″ S, 28°58′17″ E). The bacterial strains were phylogenetically identified in our previous work [20]. The bacterial cultures were maintained by streaking on nutrient agar medium and incubated at 35 °C in a static incubator for 24 h. Thereafter, the cultures were kept at 4 °C. The maintenance was done every two weeks.

### 2.2. Chemicals and Media

Butyric acid was purchased from Sigma-Aldrich Inc., St. Louis, MO, USA. HPLC grade $H_2SO_4$ was purchased from Glassworld, South Africa. NaOH, HCl, 37% W/W, as well as all other chemicals used for the preparation of the growth medium were purchased from Merck Chemicals (Pty) Ltd., Johannesburg, South Africa. Ultra-filtered, deionized water (18.2 MΩ) was prepared by a Purelab Flex purification system, ELGA Lab Water Ltd., London, U.K. Sterilized deionized water was used to make 6 M NaOH. All chemicals used in the experiments were of analytical grade—the highest purity available.

The mineral salt medium (MSM) used for the isolation, maintenance, and growth of bacteria, as well as the bacterial degradation of butyric acid, was prepared according to Roslev et al. [29]. The pH of the MSM was adjusted to 7.0 with 6 M NaOH unless otherwise stated. The other medium that was used was 31 g of nutrient agar in 1 L. Both culture media were prepared in deionized water (18.2 MΩ) and sterilized by autoclaving at a temperature of 121 °C and pressure of 115 kg/cm$^2$ for 15 min prior to use.

### 2.3. Phylogenetic Tree Construction

The phylogenetic trees of *S. marcescens* and *B. cereus* were constructed using the best-fit evolutionary model parameters determined by MEGA version 6 [30]. The best of the Nearest Neighbor Interchange and Subtree-Pruning-Regrafting search algorithms were applied for tree searching. Branch support was evaluated using bootstrap analyses based on the same model parameters and was estimated using 100 pseudo replicates [30].

### 2.4. Bacterial Consortia Development

The isolates for the consortium development were selected based on two categories. The first category comprised of three bacterial strains that were able to degrade butyric acid completely within 20 h. The second category comprised of three bacterial strains that were able to degrade butyric acid within 24 h. Bacterial consortia were formulated using the selected bacterial strains by applying a combinational statistical formula, which is denoted by Equation (1) [31]:

$$(n\ r\ ) = \frac{n!}{r!(n-r)!} \qquad (1)$$

where $(n\ r)$ is the combinatorial symbol, read as "$n$ choose $r$", $n$ is the total number of bacterial strains, and r is the number of bacterial strains in each consortium. The bacterial consortia were developed by aseptically mixing in 1/1 (V/V) (1 mL of pure bacterial cell suspension with absorbance of 2.0 ($OD_{600}$)) into a 50 mL pre-sterilized centrifuge tube (Greiner Bio-One, Kremsmünster, Austria). The mixture was then vigorously vortexed to ensure the homogenous distribution of all bacterial strains.

*2.5. Butyric acid Degradation by Pure cultures and Bacterial Consortia*

The experiments were carried out by inoculating 1 mL of bacterial consortium or pure bacterial cultures into 150 mL each of MSM supplemented with 1000 mg/L of butyric acid as a sole carbon source in a sterile 250 mL Erlenmeyer volumetric flask in triplicates. Abiotic MSM with the same butyric acid concentration was used as the control in triplicates. After sealing with aseptic cotton wool, the flasks were incubated in the dark at 30 °C in a temperature-controlled rotary shaker at an agitation rate of 110 rpm for 24 h. The samples were taken aseptically at regular 4 h time intervals to determine the butyric acid concentration, as well as the optical density. The samples for determining the butyric acid degradation were taken at time, t = 4, 8, 12, 16, 20, and 24 h. The suspensions were vortexed and centrifuged for 10 min at 10,000 rpm at 4 °C. The supernatant from each sample was analyzed by high-performance liquid chromatography (HPLC), as explained in Section 2.7. Based on Equation (2), the degradation efficiency at given time intervals was expressed as the percentage of butyric acid degraded in relation to the remaining butyric acid in appropriate abiotic control samples

$$D_t\ (\%) = ((B_i - B_t)/B_i) \times 100 \qquad (2)$$

where $D_t$ was the butyric acid degradation efficiency (%) at incubation time (h), $B_i$ was the initial butyric acid concentration (mg/L) of the abiotic sample, and $B_t$ was the butyric acid concentration (mg/L) of the biotic sample at incubation time, t (h).

*2.6. Effect of Environmental Parameters on Bacterial Growth and Butyric Acid Degradation*

The effects of temperature, pH, and inoculum size on butyric acid degradation and the growth of *S. marcescens* and *B. cereus* were investigated. Bacterial strain cell suspension was inoculated in 250 mL Erlenmeyer flasks in triplicates containing 150 mL MSM supplemented with 1000 mg/L of butyric acid as a sole source of carbon. The Erlenmeyer flasks were kept at varied temperatures, initial pH values, and initial inoculum sizes and incubated for 16 h by inoculating ($OD_{600} = 2.0$) the biomass of each of the bacterial strains separately unless otherwise stated. The effects of temperature on butyric acid degradation and bacterial growth were assessed at various temperatures of 25, 30, 35, 40, and 45 °C at pH 7 and 110 rpm. The effects of the initial pH value on butyric acid degradation and bacterial growth were assessed with MSM initial pH values of 5, 6, 7, 8, 9, and 10 at 30 °C and 110 rpm. The initial pH values were obtained by the titration of concentrated HCl or 6M NaOH. To assess the effect of the initial inoculum concentrations on butyric acid degradation and bacterial growth, MSM was inoculated with 1 mL of cell suspension with varied inoculum sizes of 0.5, 1.0, 1.5, 2.0, and 2.5 at 30 °C, pH 7, and 110 rpm. Abiotic controls were also set up for each experiment. After 16 h, the butyric acid degradation efficiencies in the respective cultures' flasks were determined based on Equation (2).

*2.7. Analytical Methods*

The degradation of butyric acid was determined by HPLC using a Waters Alliance 2695 Separation Module HPLC system (Waters Corporation, Milford, MA, USA) equipped with Aminex HPX-87H ion-exclusion organic acid, 300 mm × 7.8 mm, 9 µm particle size column (Bio-Rad Laboratories, Berkeley, CA, USA). The mobile phase was 0.02 M$H_2SO_4$ (1.1 mL of 98% $H_2SO_4$ with 18.2 MΩ

deionized water to a final volume of 1.0 L). The isocratic flow rate used was 1 mLmin$^{-1}$ and the column temperatures were maintained at 60 °C. An injection volume of 10 μL was used for all analyses. The column eluent was passed through a Waters 2998 photodiode array detector (PAD) equipped with a micro UV cell (Waters Corporation, Milford, MA, USA) monitored at a wavelength of 210 nm. The bacterial growth was spectrophotometrically monitored by measuring the optical density at single wavelength λ = 600 nm (OD$_{600}$) using a UV Lightwave II spectrophotometer (Labotec, Gauteng, South Africa).

## 3. Results and Discussion

### 3.1. Selection of the Bacterial Strains

The effective biodegradation of butyric acid entails the presence of an acclimatized microbial population capable of degrading butyric acid. Bacterial isolates were consequently obtained from composite fecal sludge from the pit latrines that had butyric acid as one of the emitted compounds responsible for malodors [17]. The detailed representative total ion chromatogram of the fecal sample is presented in Figure 1. These bacterial isolates were well adapted to the pit latrine environment; hence, it is assumed that they have better potential for the degradation of butyric acid in such similar environments. In our previous work [27], nine bacterial strains were found to possess butyric acid-degrading capabilities.

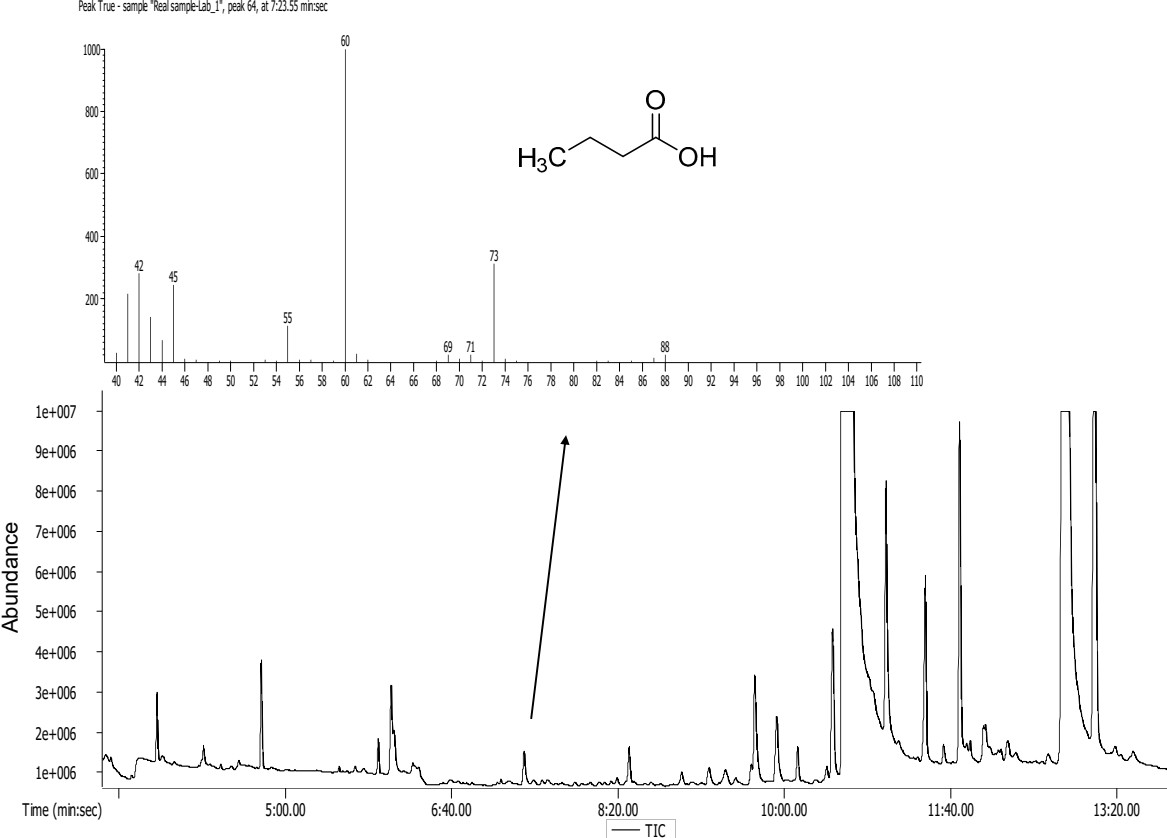

**Figure 1.** Detailed total ion chromatogram of fecal sludge sample, mass spectrum of butyric acid, and chemical structure of butyric acid.

However, in this work, only six bacterial strains were selected. These butyric acid-degrading bacterial strains were phylogenetically identified based on the 16S rRNA gene sequencing (Table 1). The bacterial strains were selected because they are commonly associated with fecal sludge according to previous studies found in the literature. For instance, in their study on the estimation of nitrous

oxide ($N_2O$) release from pit toilets in Mulbagal town, Karnataka, India, Rao et al. [32] revealed that the microbial denitrification reaction was facilitated by *Pseudomonas* spp., *Serratia* spp., *B. cereus*, *B. subtilis*, and *Achrobacter* spp. Similarly, Déportes et al. [33] and Carrington [34] have indicated that *Bacillus* spp., *Pseudomonas* spp., and *Serratia* spp. were bacteria of epidemiological concern that are associated with fecal sludge. Thus, the isolation of the indigenous bacterial strains that are acclimated to the local environmental conditions of pit latrine fecal sludge are vital for the microbial proficiency of butyric acid removal in the pit latrine or analogous environments [35].

**Table 1.** Computation of Bacterial Consortia Using the Six Selected Butyric Acid-Degrading Bacteria: *A. xylosoxidans* (AX), *B. cereus* (BC), *P. aeruginosa* (PA), *S. marcescens* (SM), *A. animicus* (AA), and *Alcaligenes* sp. Strain SY1 (AS).

| Bacterial Strains Group | Computed Bacterial Consortia Using Combinations | Consortium Designation |
|---|---|---|
| Top Two Best Bacterial Strains in Category 1 and Best Bacterial Strain in Category 2 [AX, SM, BC] | [AX, SM]<br>[AX, BC]<br>[SM, BC] | C1<br>C2<br>C3 |
| Top Two Best Bacterial Strains in Category 1 and Top Two Best Bacterial Strains in Category 2 [AX, SM, BC, AA] | [AX, SM, BC]<br>[AX, SM, AA]<br>[AX, BC, AA]<br>[SM, BC, AA] | C4<br>C5<br>C6<br>C7 |
| All Three Bacterial Strains in Category 1 and Top Two Best Bacterial Strains in Category 2 [AX, SM, BC, AA, PA] | [AX, SM, BC, AA]<br>[AX, SM, BC, PA]<br>[AX, SM, AA, PA]<br>[AX, BC, AA, PA]<br>[SM, BC, AA, PA] | C8<br>C9<br>C10<br>C11<br>C12 |
| All Three Bacterial Strains in Category 1 and all Three Bacterial Strains in Category 2 [AX, SM, BC, AA, PA, AS] | [AX, SM, BC, AA, PA]<br>[AX, SM, BC, AA, AS]<br>[AX, SM, BC, PA, AS]<br>[AX, SM, BC, PA, AS]<br>[AX, BC, AA, PA, AS]<br>[SM, BC, AA, PA, AS] | C13<br>C14<br>C15<br>C16<br>C17<br>C18 |
| All Six Bacterial strains | [AX, SM, BC, AA, PA, AS] | C19 |

### 3.2. Butyric Acid Degradation by Pure Bacterial Strains

To take the role as butyric acid attenuation agents, the bacterial strains ought to have the capacity to grow in an environment that contains a high concentration of butyric acid as a sole source of carbon. Butyric acid degradation by individual pure bacterial strains was assessed at an initial butyric acid concentration of 1000 mg/L in a defined MSM. The choice of 1000 mg/L butyric acid concentration in the present study was based on the butyric acid concentration used for experiments in our previous study [27]. According to Lin et al. [17], 90% of pit latrines surveyed in Durban, Nairobi, Kampala, and Pune, and the model toilet had a butyric acid concentration between 46.2 and 1042 mg/L. The comparison of butyric acid degradation efficiencies by the individual bacterial strain cultures is shown in Figure 2a, and their corresponding bacterial growth curves are indicated in Figure 2b. The results showed that butyric acid biodegradation occurred in each of the bacterial strains, as measured by the HPLC analysis and comparison to abiotic controls.

The butyric acid degradation efficiency results reflected the relationship between bacterial growth and butyric acid degradation. In all the experiments, the removal of butyric acid was accompanied by a concomitant increase in bacterial growth even though the length of the lag phase varied between the bacterial strains. After 4 h, the results showed that *A. xylosoxidans* degraded 4.0% of the butyric acid and *B. cereus* degraded 3.7% of the butyric acid when compared with the abiotic control. At the end of 8 h, *B. cereus*, *P. aeruginosa*, *A. xylosoxidans*, *A. animicus*, *S. marcescens*, and *Alcaligenes* spp. strain SY1 degraded 24.9%, 12.16%, 7.03%, 4.84%, 3.94%, and 2.10%, respectively, when compared with the abiotic control. The results suggested that the inoculum degradation efficiencies of butyric acid past 8 h incubation time were in descending order, as follows: *A. xylosoxidans* > *B. cereus* >

*P. aeruginosa* > *S. marcescens* > *A. animicus* > *Alcaligenes* spp. strain SY1 compared to the abiotic control. However, it is also evident from Figure 2a that each bacterial strain showed variations in butyric acid degradation efficiencies at different times. It was observed that the rates of degradation were high in the exponential phase of bacterial growth. It was clearly shown that all three bacterial strains—*P. aeruginosa*, *A. xylosoxidans*, and *B. cereus* could perform the complete degradation of 1000 mg/L butyric acid within 20 h of incubation, as shown in Figure 2a.

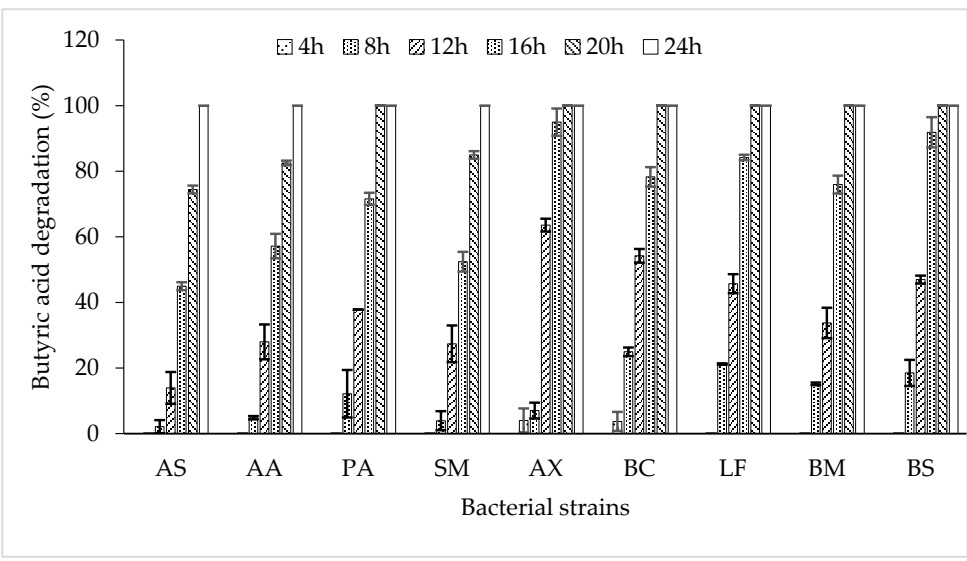

(**a**)

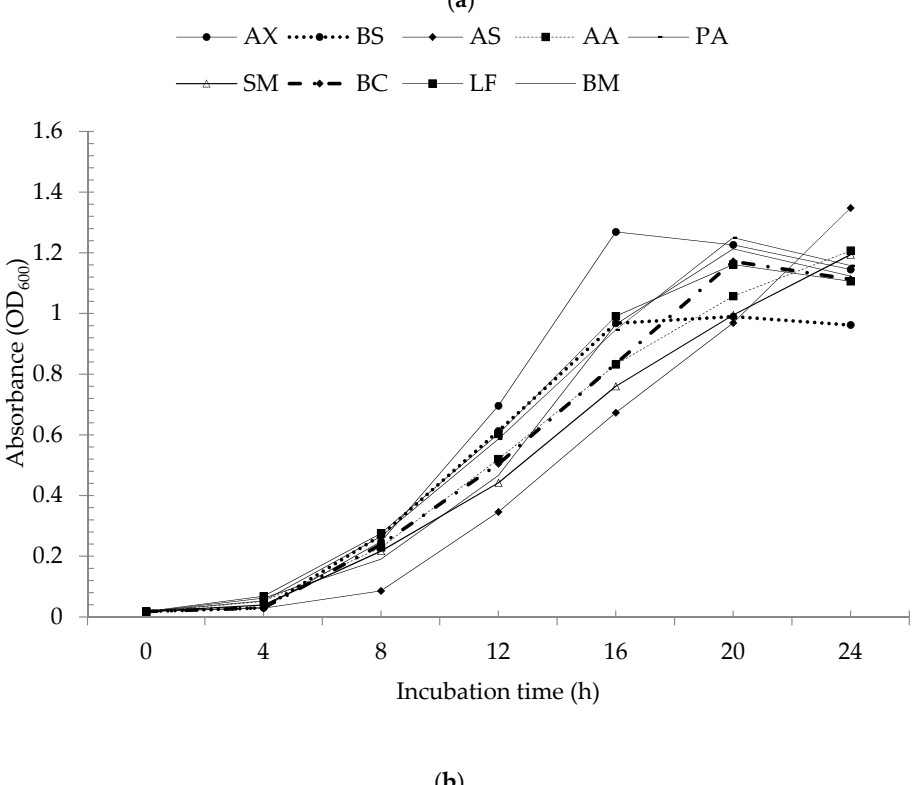

(**b**)

**Figure 2.** (**a**) Butyric acid degradation efficiencies by different bacterial strains and (**b**) growth kinetics of different bacterial strains.

Whilst the bacterial strains *Alcaligenes* sp. strain SY1, *A. animicus*, and *S. marcescens* could degrade 1000 mg/L of butyric acid completely within 24 h of incubation, no butyric acid degradation was observed in the abiotic controls. Studies regarding bacteria-degrading butyric acid have been reported in the literature [24–26]. Nevertheless, except for the study by Chin et al. [26], all these studies were performed with butyric acid as a sole source of carbon. To the best of our knowledge and after a thorough literature search, none of the bacterial strains used in this work have been specifically reported as butyric acid-degrading bacteria.

### 3.3. Degradation of Butyric Acid by Bacterial Consortia

Characteristically, the application of individual bacterial strains for biodegradation does not represent the real situation of environmental microorganisms during the biodegradation of butyric acid in pit latrine fecal sludge. This is because in real environmental settings, biodegradation relies on the cooperative metabolic activities of mixed microbial populations. There are no studies in the literature regarding the construction of bacterial consortia from bacterial strains isolated directly from pit latrine fecal sludge for the biodegradation of butyric acid. Nineteen different bacterial consortia that were constructed involving the selected bacterial strains in five different combinations are presented in Table 1.

The successful bacterial consortium was established based on the compatibility of the individual component bacterial strains of the consortium. Hence, there was an absence of any antagonism among constituent bacterial strains to concomitantly accomplish all the metabolic processes for enhanced degradation [36]. From the 19 constructed bacterial consortia, the best performing bacterial consortia were selected based on comparatively higher butyric degradation efficiency in relation to the mean degradation efficiencies of the individual component bacterial strains studied after 16 h of incubation under the same environmental conditions. The 16-h incubation time was chosen for comparison because some of the bacterial consortia had already achieved 100% butyric acid degradation within 16 h of incubation. Butyric acid degradation, which was examined after 16 h of incubation, as measured by the HPLC analysis, is shown in Figure 3. The butyric acid degrading bacteria in all the samples achieved the degradation efficiencies in the range of 55.6% to 100% after 16 h of incubation, as monitored by the HPLC.

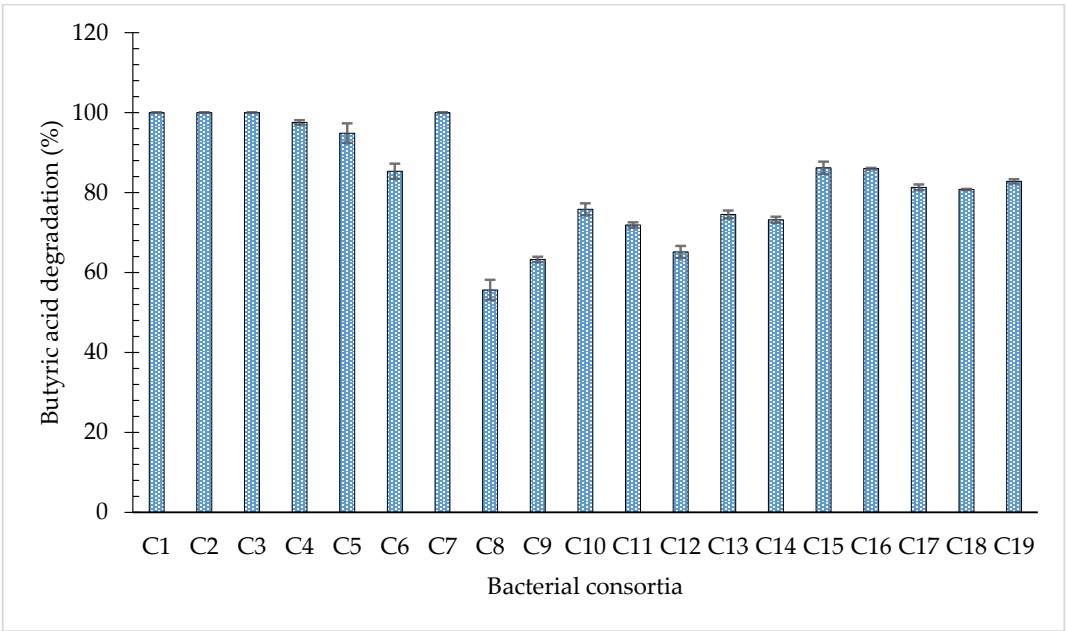

**Figure 3.** Butyric acid degradation by different constructed bacterial consortia after 16 h of incubation at pH 7, 30 °C, and 110 rpm.

The concentration of butyric acid in all the treatments varied throughout the incubation period monitored (data not shown). The samples inoculated with bacterial consortia, C1, C2, and C3, achieved 100% butyric acid degradation within 16 h of incubation. However, the bacterial consortium, C3 (combination of *S. marcescens* and *B. cereus*), had the highest degradation efficiencies at all sampling times (4 h, 8 h, and 12 h) compared to the other bacterial consortia that degraded butyric acid completely within 16 h (data not shown). Moreover, the butyric acid degradation efficiency of consortium, C3, was higher than the individual butyric acid degradation efficiencies of *S. marcescens* and *B. cereus* of 52.4% and 78.3%, respectively. This is much higher than the butyric acid degradation efficiencies of the individual component bacterial strains of consortia, C1 and C2, as shown in Table 2. The phylogenetic trees of *S. marcescens* and *B. cereus*, showing the closest relatives as maintained by National Centre for Biotechnology Information using BLAST searches at http://www.ncbi.nlm.nih.gov/BLAST based on the 16S rRNA gene sequence are shown in Figure 4a,b, respectively. Substantial degradation efficiencies of butyric acid of 86.0%, 99.9%, and 97.6% were also achieved in the samples inoculated with bacterial consortia C4, C7, and C16, respectively. The butyric acid degradation efficiencies of C7 and C16 were the same. This was also higher compared to the butyric acid degradation efficiencies of the individual component bacterial strains. It was apparent that the biodegradation of butyric acid by these consortia were more effective, as they outperformed the individual component bacterial strains of the consortia.

**Table 2.** Degradation Efficiencies of Individual Bacterial Strains and Their Mean Efficiencies and Degradation Efficiencies of Consortia.

| Degradation Efficiencies of Individual Bacterial Strains (%) | | | | | | Mean Efficiency (%) | Consortium | Degradation Efficiencies of Consortia (%) |
|------|------|------|------|------|------|------|------|------|
| AX | SM | BC | AA | PA | AS | | | |
| 94.97 | 52.41 | - | - | - | - | 73.69 | C1 | 100 |
| 94.97 | - | 78.29 | - | - | - | 86.63 | C2 | 100 |
| - | 52.41 | 78.29 | - | - | - | 65.35 | C3 | 100 |
| 94.97 | 52.41 | 78.29 | - | - | - | 75.22 | C4 | 97.56 |
| 94.97 | 52.41 | - | 57.16 | - | - | 68.18 | C5 | 94.86 |
| 94.97 | - | 78.29 | 57.16 | - | - | 76.81 | C6 | 85.33 |
| - | 52.41 | 78.29 | 57.16 | - | - | 62.62 | C7 | 99.89 |
| 94.97 | 52.41 | 78.29 | 57.16 | - | - | 70.71 | C8 | 55.66 |
| 94.97 | 52.41 | 78.29 | - | 71.57 | - | 74.31 | C9 | 63.28 |
| 94.97 | 52.41 | - | 57.16 | 71.57 | - | 69.03 | C10 | 75.82 |
| 94.97 | - | 78.29 | 57.16 | 71.57 | - | 76.24 | C11 | 71.89 |
| - | 52.41 | 78.29 | 57.16 | 71.57 | - | 64.85 | C12 | 65.15 |
| 94.97 | 52.41 | 78.29 | 57.16 | 71.57 | - | 70.88 | C13 | 74.53 |
| 94.97 | 52.41 | 78.29 | 57.16 | - | 44.94 | 65.55 | C14 | 73.21 |
| 94.97 | 52.41 | 78.29 | - | 71.57 | 44.94 | 68.44 | C15 | 86.21 |
| - | 52.41 | 78.29 | 57.16 | 71.57 | 44.94 | 60.87 | C16 | 86.01 |
| 94.97 | - | 78.29 | 57.16 | 71.57 | 44.94 | 69.39 | C17 | 81.31 |
| 94.97 | 52.41 | - | 57.16 | 71.57 | 44.94 | 64.21 | C18 | 80.79 |
| 94.97 | 52.41 | 78.29 | 57.16 | 71.57 | 44.94 | 67.39 | C19 | 82.82 |

The results suggested that bacterial synergism may be indispensable for butyric acid degradation in the pit latrine fecal sludge where the bacterial strains were isolated. It is widely recognized in the field of microbiology that coordinated bacterial consortia have the potential to be more productive, robust, and effective to environmental fluctuations than individual pure bacterial cultures [37]. This is undoubtedly because of the concerted activities of the individual component bacterial strains of the consortium.

The interspecific interactions within the constructed bacterial consortia that coxswained to improve the degradation ability were not determined. However, numerous mechanisms that promote synergetic interactions between constituent members of the degradative mixed communities in nature have been postulated by Deng and Wang [38], but in the present study, two possible mechanisms may be offered at this point based on previous reports inter alia, including:

(i)　the metabolic and physiological inadequacies of one bacterial strain in the consortium are compensated for by the presence of other bacterial strains in the consortium with the appropriate

complementary physiology, which are able to provide the appropriate metabolic benefit to all
bacterial strains involved [39];

(ii) associated metabolism, wherein one bacterial strain in the consortium take up the intermediates
of the metabolic pathway released during the degradation of butyric acid, which may be toxic
and which may hinder the metabolic activities, thus, appearing to protect the other constituents
of the bacterial consortium from toxicity that would otherwise accrue from the accumulation of
the metabolites [40].

(**a**)

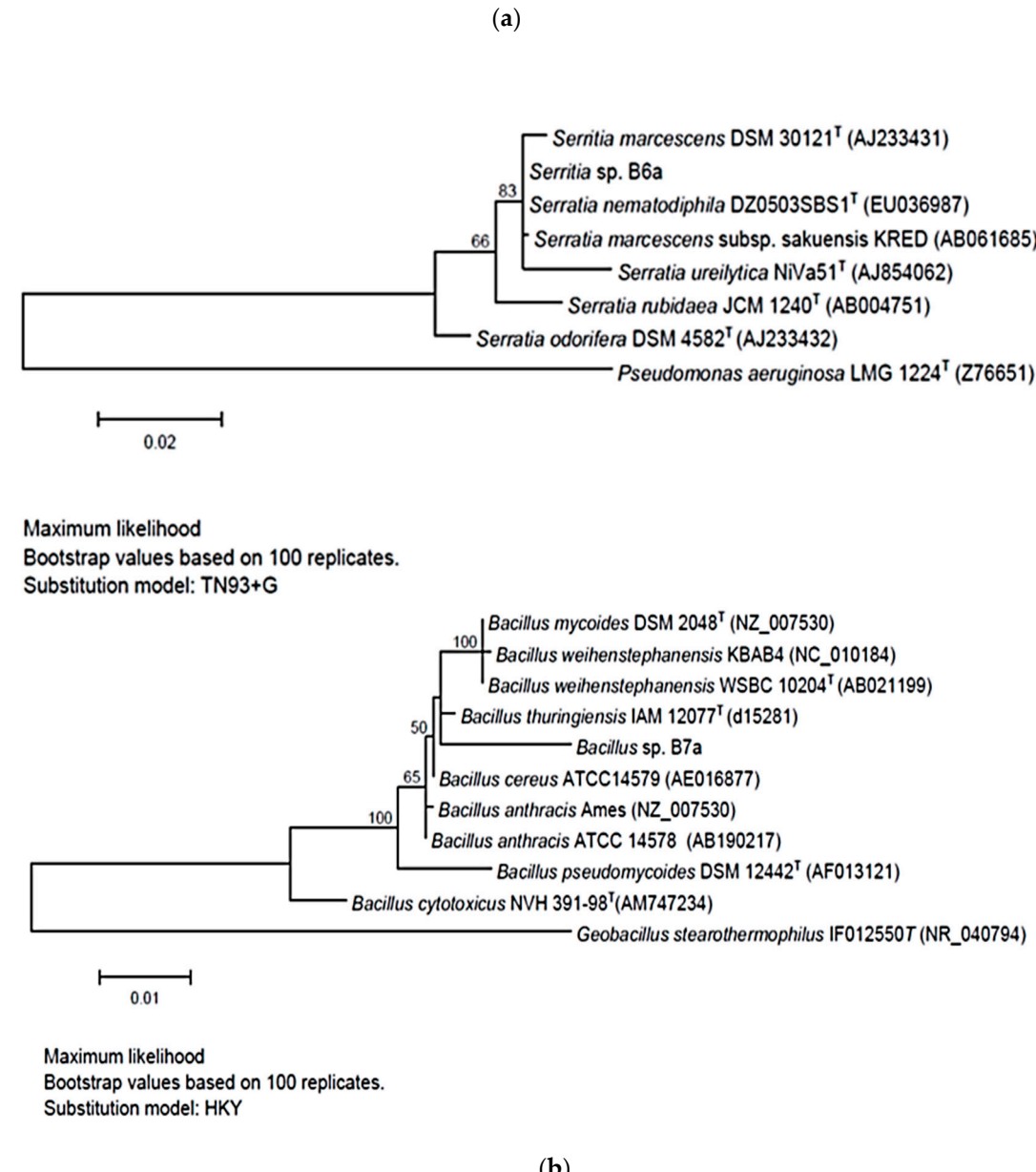

(**b**)

**Figure 4.** Phylogenetic tree for (**a**) *S. marcescens* and (**b**) *B. cereus* and related strains based on 16s rRNA
gene sequences. Bootstrap values were based on 100 replicates.

Several studies of the biodegradation potency of bacterial consortia exhibiting similar results
were reported earlier. Thus, for instance, a defined consortium of indigenous *Pseudomonas*
and actinobacteria offered a synergetic activity for effective poly aromatic hydrocarbon removal
capabilities when compared to their pure cultures [41]. A mixed bacterial consortium described by
Sathishkumar et al. [42], in which *Bacillus* sp. IOS17, *Corynobacterium* sp. BPS2-6, *Pseudomonas* sp.

HPS2-5, and *Pseudomonas* sp. BPS1-8 incubated together showed the superior growth and degradation of crude oil to individual bacterial strains. Saratale et al. [43] reported evidently higher degradation and decolorization efficiency for a mixture of reactive dyes by a bacterial consortium of *Proteus vulgaris* and *Micrococcus glutamicus* compared to the use of individual bacterial strains. Similarly, Tizntzun-Camacho et al. [44] found low hexadecane degradation efficiencies by pure cultures of *Xanthomonas* sp., *Acinobacter bouvetti*, and *Defluvibacter lusatiensis*, which were noticeably enhanced (79 ± 3%) when such bacterial strains were grown together. The concerted metabolic potential of the mixed cultures to degrade butyric acid has also been reported by Kristiansen et al. [45], wherein uncharacterized bacterial strains only identified as members of phyla—*Microbacterium*, *Gordonia*, *Acetobacteria*, *Rhodococus*, *Propionibacteria*, *Janibacter*, *Alpha-*, *Beta-*, and *Gamma*—proteobacteria were used. The results showed up to 70% reduction of organic acids, including butyric acid, in the presence of other odorous compounds in a full-scale biological air filter treating air from a pig facility. Similarly, Sheridan et al. [14] used a mixed aerobic microbial culture consisting of two fungi and five bacterial strains of phylum, Gamma-proteobacteria, identified as members of genera—*Moraxella*, *Enterobacter*, and *Pseudomonas*—which were isolated from under a diesel storage tank for the degradation of butyric acid from waste exhaust air.

Other bacterial consortia—C5, C6, C8, C9, C10, C11, C12, C13, C14, C15, C17, C18, and C19—exhibited lower degradation efficiencies compared to the degradation efficiencies of at least one of the individual component bacterial strains that made up each of them, as shown in Table 2. The possible explanation could be that the bacterial strains were engaged in competition for a pool of resources with limited availability, such as space, dissolved oxygen, and nutrients. This is very common for constituent bacterial strains with analogous nutritional requirements within the consortium [46,47]. Studies that have demonstrated that the combined efforts of consortia may not always have a synergetic effect for all the substrates were also found in the literature. Kumar and Phillip [48] reported that the degradation of endosulfan was performed potently better in monocultures of three bacterial strains—*Staphylococcus* sp., *Bacillus circulans*-I, and *Bacillus circulans*-II—than by them in a consortium. Similar results were observed by Guo et al. [49], who demonstrated that the isolate of *Paracoccus* spp. was more efficient in the degradation of pyrene than that of the mixed cultures.

### 3.4. Environmental Factors Affecting the Growth and Butyric Acid Biodegradation by Bacterial Consortium, C3

Effective biodegradation can only be achieved when environmental conditions are favorable for microorganisms' metabolic activities [50]. In the present study, factors such as temperature, pH, and inoculum size, as explained in Section 2.6, were considered for each of the bacterial strains (*B. cereus* and *S. marcescens*) of bacterial consortium, C3, to elucidate how they can affect the accomplishment of the butyric acid biodegradation process and their growth.

#### 3.4.1. Effect of Incubation Temperature

In microbiology, it is well established that biological processes, such as aerobic metabolism and growth, are known to exhibit environmental temperature dependence [51]. Temperature influences the rates of enzymatically catalyzed reactions and the diffusion rate of the substrate to the cell [52]. Both *S. marcescens* and *B. cereus* degraded significant quantities of butyric acid at various incubation temperatures (25, 30, 35, and 40 °C) at 16 h of incubation, as shown in Figure 5. The results show that the growth and butyric acid degradation of both *S. marcescens* and *B. cereus* were optimal at an incubation temperature of 30 °C. It was observed that there was a slight gradual decrease in the bacterial growth and degradation efficiencies of butyric acid when the incubation temperature decreased from 30 to 25 °C. The degradation efficiencies of *S. marcescens* and *B. cereus* decreased from 72.41% to 70.42% and 78.29% to 66.37%, respectively, at 16 h of incubation. This could be due to a reduced catalytic capacity at lower temperatures [51]. However, the bacterial growth and butyric acid degradation efficiencies decreased when the incubation temperature increased by the same 5 °C with comparatively higher decreased degradation efficiencies at 16 h of incubation. This suggests the bacterial strains

were much less sensitive to low temperatures than high temperatures. There was a drastic decrease in bacterial growth and butyric acid degradation efficiencies at 16 h of incubation with an increase of incubation temperature from 40 to 45 °C. This could be attributed to the denaturation of proteins at high temperatures. This is because with a further rise in temperature, the components with heat sensitivity, such as enzymes, which are secreted outside the cell into the surrounding medium to perform metabolic processes, are irreversibly denatured and growth rates drop quickly and cause inhibition and then mortality [53]. Furthermore, with increasing temperature, the solubility of oxygen is decreased in the aqueous phase and, as a result, the metabolic activity of aerobic microbes is reduced [54].

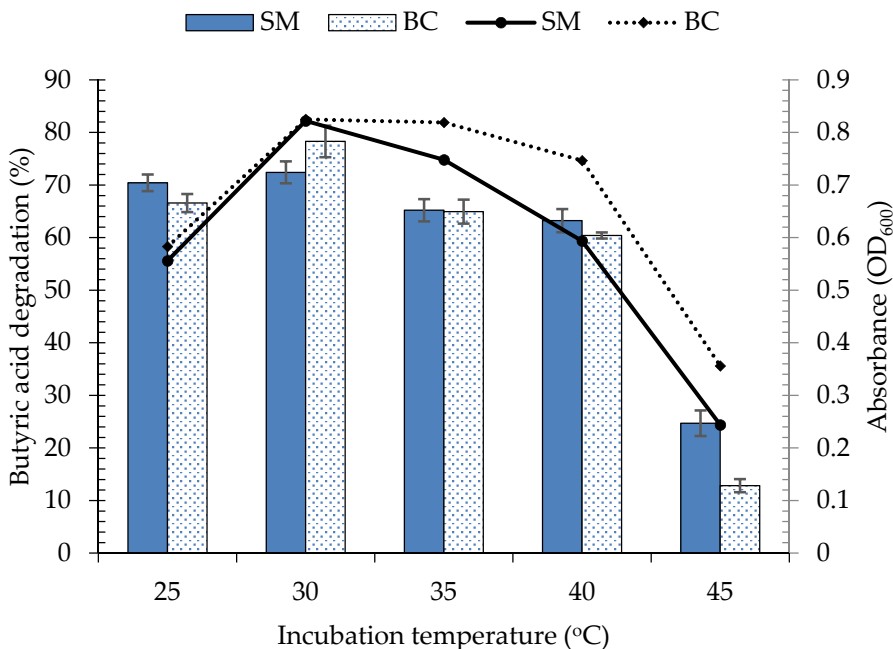

**Figure 5.** Growth and degradation of butyric acid after 16 h of inoculation by *S. marcescens* and *B. cereus* at different incubation temperatures: butyric acid degradation efficiency: bar graphs; bacterial growth; line graph.

This is supported by the previous report that temperatures higher than the organism's optimum temperature range causes cell death, which is fast, while lower temperatures still result in cell death rate, which is slower [55]. The complete degradation of butyric acid was observed in the inoculated flasks incubated at 30 °C at 16 h of incubation. These results are in accordance with the work of Chin et al. [26], who reported that a temperature of 30 °C was an optimal incubation temperature for the degradation of the butyric acid by *Acinetobacter calcoaceticus* at a pH of 7 under aerobic conditions. The literature available regarding temperature values inside pit latrines is limited. However, a study by Sherpa et al. [56] found that the temperature of fecal sludge sampled from urine-diverting dehydrating toilets with ash as a primary additive in Kathmandu Valley, Nepal was in a range of 19.5–32.8 °C. Similarly, Nabateesa et al. [57] investigated the temperature of fecal sludge inside pit latrines in Kampala, Uganda. The temperature was found to be in a range of 22.3–30.7 °C, with an overall mean of 25.4 °C, with higher temperatures in the top layer and decreasing with depth. According to Nwaneri et al. [58], aerobic processes inside pit latrines occur in the top layers of fecal sludge portions of pit contents. Therefore, the temperature and the aerobic nature of fecal sludge in the top layer of pit contents can favorably support the metabolic activities of the bacterial strains, since it provides the mesophilic temperature range at which both *S. marcescens* and *B. cereus* optimally grow and degrade butyric acid.

### 3.4.2. Effect of Initial pH of the Medium

pH is another important parameter for microbes and different species prefer different pH values. Environmental pH has a strong effect on their cell metabolism and growth. The effect of medium pH on the bacterial growth and butyric acid degradation efficiencies over an initial medium pH of 5 to 10 are shown in Figure 6. The pH range was carefully chosen to mimic the range of pH values found in pit latrine fecal sludge in the range of environmental settings according to previous studies [57,59,60]. However, the pH values of fecal sludge are more complex, since they are influenced by numerous factors [60].

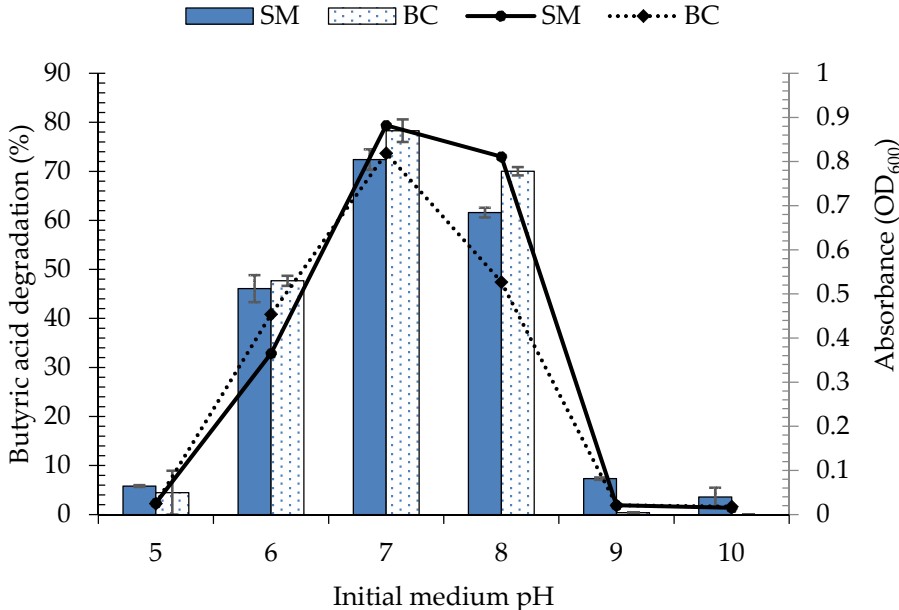

**Figure 6.** Growth and degradation of butyric acid after 16 h of inoculation by *S. marcescens* and *B. cereus* at different initial medium pH: butyric acid degradation efficiency: bar graphs; bacterial growth: line graph.

In the samples inoculated with *S. marcescens* or *B. cereus*, substantial bacterial growth and butyric acid degradation were observed at an initial pH of 5 to 7 at 16 h of incubation. An increase in the initial pH above pH 7 significantly decreased the bacterial growth butyric acid degradation of both bacterial strains. In the range of initial pH investigated, the highest bacterial growth and butyric acid degradation for both *B. cereus* and *S. marcescens* were achieved at an initial medium pH of 7 at 16 h of incubation. This suggests that the butyric acid-degrading enzymes have their optimal enzymatic activity in neutral surroundings, implying that the bacterial strains are neutrophiles. Coincidentally, the optimal degradation of butyric acid as the sole carbon source inoculated with *Acinetobacter calcoaceticus*, *Burkholdeira cepacia*, and *Wautersia paucula* was accomplished at pH 7.0 [26].

In the pH values outside the range of 6 to 8, the bacterial strains exhibited a characteristic sensitivity to pH that inhibited bacterial growth and butyric acid degradation. It is, however, worth mentioning that the bacterial strains might have mechanisms to modify the pH of the medium. It was noted that the pH of the culture medium with initial pH values in the acidic condition increased with incubation time and shifted towards the optimal pH. On the other hand, the pH of the culture medium with initial pH values of the extreme alkaline condition decreased with incubation time and shifted towards the optimal pH (data not shown). The increasing and lowering of pH of the culture medium could be due to the production of organic acids and ammonia, respectively, as metabolic products [26,61]. However, further research is required to understand the mechanisms that the bacterial strains use to modify the extremes of medium initial pH.

### 3.4.3. Effect of Initial Inoculum Size

To ascertain the effect of the initial inoculation size on the degradation of butyric acid and bacterial growth, the initial inoculum sizes varied from 0.5 to 2.5. Only 1 mL in a volume of cell suspension prepared with these optical densities was used. This means that different inoculum sizes affected the initial population of bacteria in the medium. After increase in the initial inoculum sizes of *B. cereus* from 0.5 to 2.0 after 16 h of incubation, the bacterial growth increased marginally and the butyric acid degradation efficiencies varied, but not significantly, as shown in Figure 7. The optimal degradation efficiency was reached at 2.0. However, with an increase in inoculation concentration above 2.0, there was a decrease in butyric acid degradation, as well as bacterial growth. The decrease in degradation efficiency with a further increase in initial inoculum size is not a new phenomenon. Increasing the inoculum size of *Bacillus thuringiensis* did not result in the dimethyl phthalate degradation [62]. This is because a bacterial population rise also intensifies the bacterial competition for resources, such as substrates, oxygen, space, etc., and, therefore, restricts bacterial growth when these resources are depleted in the medium [63].

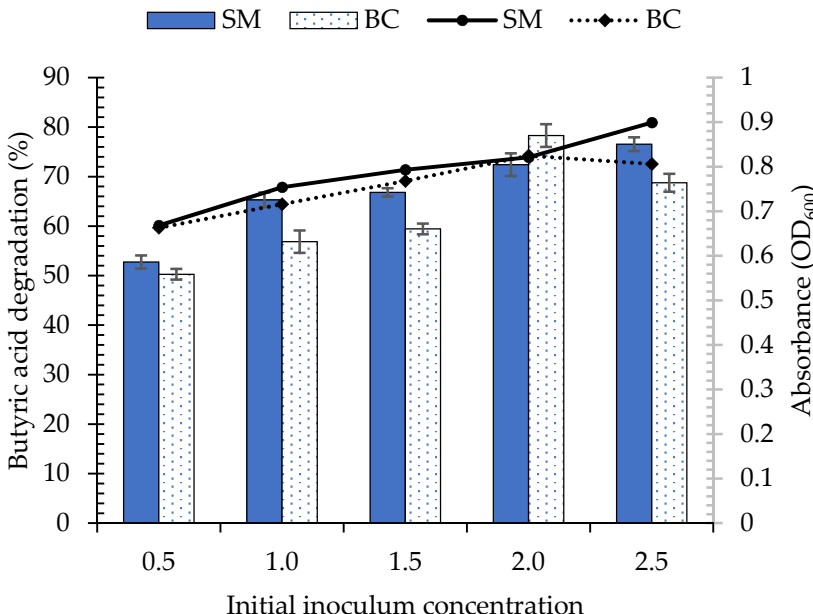

**Figure 7.** Growth and degradation of butyric acid after 16 h of inoculation *by S. marcescens* and *B. cereus* with different initial inoculum sizes: butyric acid degradation efficiency: bar graphs; bacterial growth: line graph.

The results showed that an increase in inoculum size from 0.5 to 1.5 did not reduce the lag phase (data not shown) to help the batch system to attain the exponential growth phase rapidly to attain significant butyric acid degradation. This could be because the initial densities of the bacteria were not large enough to ensure quick proliferation and biomass synthesis in the cultivation [64], while for *S. marcescens*, the optimum degradation was attained with an inoculum size of 2.5. For economic and comparison purposes, the inoculum size was not increased. The increase in the initial inoculum size of *S. marcescens* resulted in increased butyric acid degradation and bacterial growth. Bildan and Monomania [65] reported that the aerobic degradation of dichlorodiphenyltrichloroethane by *S. marcescens* DT-1P increased with an increase in inoculum size in liquid culture. The differences between the two bacterial strains could be attributed to the fact that dissimilar bacterial strains have different population sizes that can do rapid and complete butyric acid degradation.

## 4. Conclusions

This work provides an insight into the potential impacts of the composition of bacterial consortium on the biodegradation of butyric acid. The work has shown that the effectiveness of the constructed bacterial consortia to enhance butyric acid degradation is not simply a result of the adding together of the individual component bacterial strain degradation capacities of the consortium. The co-culture of *S. marcescens* and *B. cereus* was selected as an effective butyric acid degrading consortium that degraded 1000 mg/L butyric acid in liquid culture within 16 h. This may be the first instance in which 1000 mg/L of butyric acid degradation has been achieved in a short incubation time of 16 h. However, the detailed molecular mechanism of butyric acid degradation by consortium C3 and the contribution of *S. marcescens* and *B. cereus* individually in the degradation process will be further carried out to envision the role that each play in the consortium. A temperature of 30 °C and pH 7 were found to be optimum for maximum degradation for both *S. marcescens* and *B. cereus*. The inoculation concentrations of 2.0 and 2.5 were optimal for maximum degradation for *S. marcescens*, *B. cereus*, and *S. marcescens*, respectively. Even though laboratory studies may not accurately reflect butyric acid biodegradation occurring in situ, this work suggests that the biodegradation process studied here has potential application for the attenuation of butyric acid-related malodors emanating from pit latrines, and this might lead to the development of better deodorization technologies. This work is of international value, as it will contribute to knowledge and progress on the bioremediation of odors emitted from pit latrine fecal sludge, hence, leading to improved sanitation uptake in developing countries.

**Author Contributions:** Conceptualization, J.B.J.N. and E.M.N.C.; methodology, J.B.J.N.; formal analysis, J.B.J.N. and R.L.S.; investigation, J.B.J.N. and R.L.S.; resources, E.M.N.C.; writing—original draft preparation, J.B.J.N. and R.L.S.; writing—review and editing, E.M.N.C.; supervision, E.M.C.N.; project administration, J.B.J.N.; funding acquisition, E.M.N.C. All authors have read and agreed to the published version of the manuscript.

**Funding:** This research was co-supported by the National Research Foundation (South Africa (NRF) Competitive Programme for Rated Researchers Grant No. CSUR180215313534 awarded to Evans M.N. Chirwa of the University of Pretoria. John Njala'mmano was supported by a postgraduate scholarship through the UP-Commonwealth programme via the Department of Research and Innovation at the University of Pretoria.

**Acknowledgments:** We would like to thank Fenus Venter for his technical assistance in the molecular identification of the bacterial strains.

**Conflicts of Interest:** The authors declare that they have no conflicts of interest.

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
