# Peer review of "In Vitro Study of Butyric Acid Deodorization Potential by Indigenously Constructed Bacterial Consortia and Pure Cultures from Pit Latrine Fecal Sludge"

_sustainability, doi:10.3390/su12125156_

Round 1

Reviewer 1 Report

Very actual topic, because approximately 946 million people are always without access to improved sanitation, practice open defecation. Even though pit latrines are the lowest rung on the sanitation ladder, they are utilised by an estimated 1.77 billion people worldwide.

There are no studies in the literature regarding the construction of bacterial consortia from bacterial strains isolated directly from pit latrine faucal sludge for biodegradation of butyric acid and the authors of this paper suggests that the bioremediation process studied here has potential application for attenuation of butyric acid related malodours emanating from pit latrines and this might lead to the development of better deodorization technologies.

The work is of international value, as it will contribute to knowledge and progress on the bioremediation of odours emitted from pit latrine faucal sludge, hence leading to improved sanitation uptake in developing countries.

I have only a few comments for improving:

Fig. 2: Why is used the blue color in the legend for 0 hours value?

In the text is written Fig. 2 a) and Fig. 2 b) but below the pictures is only two times Fig. 2; the same with the Fig.4

Is it better, if the title of the Figure is on the same page; the same for the Table 1 - not split the content

- line 412 incorrect indent.

Author Response

Dear Sir or Madam,

Kind regards,

John

Reviewer 2 Report

Dear Authors

The introduction addresses sociological, economic lines (31-41 and 62-68) and structural (lines 42-54) very interesting and necessary.
On the other hand there is a description of butyric acid (lines 55-61) and of the microorganisms and consortia (lines 69-78).
But it is necessary to explain and cite the origin of butyric acid in latrines, the generation mechanisms, environmental conditions that potentiate or inhibit it.
On the other hand, describe the mechanisms or cite experiences of biodegradation of butyric acid.

line 182, it seems to me that the expression is not "the effective bioremediation of butyric acid", it should be the effective biodegradation of butyric acid, in general the use of the bioremediation concept should be reread and rewritten, since butyric acid is not bioremediated, it biodegrades.

Section 3.4 lacks a methodological explanation that explains and relates the environmental factors considered with the respective strains. It goes immediately to a description of the responses obtained.
The most effective bacterial consortium strains are written, which is not specified and mentioned.

The phrase "bioremediation of butyric acid" is emphasized, it is the bioremediation of the latrine residue through the biodegradation of butyric acid (line 355), which is that it negatively impacts the environment.

Figures 5,6 and 7 must be improved, they do not explicitly graph what is intended to be shown, the labels must be distinguished from each other, the SM and BC are duplicated, and each pair have different meanings.
The above complicates the understanding of the associated paragraphs, the temperature has more to do with the graph and provides more complete information.
The two paragraphs associated with the remaining graphics (6 and 7) can be improved.

The conclusions are supported by the data provided by the work carried out, if a point is necessary that allows a better understanding of the results.

Author Response

Dear Sir or Madam,

Kind regards,

John

Reviewer 3 Report

General comments

  1. Introduction, Materials/Methods, Results: When you refer for a second time in the same microbial species, you should write only the first letter and not the full name of the genus. For example, the 1st time you should write Pseudomonas aeruginosa and the 2nd time write P. aeruginosa. Please correct all the species through the whole manuscript.
  2. You have used different symbols for Celsius degrees through the manuscript (e.g. lines 101, 115, 139). Please use only one symbol through the whole manuscript.
  3. Sometimes you use a thousands separator (e.g. line 143) and others you don't (e.g. line 136). You should choose only one way to express thousands through the whole manuscript.
  4. Please delete the punctuation mark comma "," before the word "and" through the whole manuscript.
  5. Add the numbers of references before the punctuation marks comma "," (e.g. line 188) and full stop "." (e.g. line 353) through the whole manuscript.

Specific comments

-Line 16: You should not use abbreviation at this point since you are not using the term MSM in the rest of the abstract.

-Line 19: As you will see in my comments in the "Results and Discussion" section (lines 277-278, 288, 350, 354), other consortiums were also effective at the same level (100% degradation). After revising the manuscript, rewrite the sentence accordingly.

-Line 21: According to the results, the optimum temperature was "30 oC" and not "35 oC". Please correct it.

-Line 22: The inoculations sizes should have a unit. Please mention the unit after the numbers.

-Line 25: What do you mean with the word "user"? Please be more specific to improve clarity.

-Lines 27-28: I would suggest you write "Serratia marcescens" and "Bacillus cereus" as keywords instead of "odour treatment" and "smell". 

-Line 31: Please delete the abbreviation "UN". It's unnecessary since you are not using it in the rest of the manuscript.

-Line 32: Please do not use the abbreviation "MDG" since you are using it only in this line of the manuscript.

-Line 33: Please write the full name of "SDG" abbreviation the first time you are referring to it in the text. Does it mean "Sustainable Development Goals"? Please explain.

-Line 51: Is it "SDG 6" or "SDG 6.2"?

-Line 61: Please rewrite the phrase. The English language needs improvement.

-Line 84: Please delete the word "strains". It's unnecessary.

-Lines 91-93: Please avoid the use of numbers (i.e. 1, 2) again. Rewrite the respective text in a better way.

-Line 105: Please write the numbers 2 and 4 of the H2SO4 formula in subscript.

-Line 106: Please write "(NaOH) and hydrochloric acid (HCl, 37% w/w)" instead of "(NaOH), 32% HCl used".

-Lines 111-112: Please write "for the isolation, maintenance and growth of bacteria as well as for the bacterial degradation of butyric acid" instead of "for isolation, maintenance, growth, and degradation of butyric acid analysis by bacteria".

-Line 114: Please leave space between "1" and "L".

-Line 119: Please delete the abbreviations "NNI" and "SPR". They are unnecessary since you are not using them in the rest of the manuscript.

-Line 131: Please write "1/1 (v/v)" instead of "1:1(v/v)". You should use the same division sign symbol (i.e. "/") through the whole manuscript.

-Line 136: Write "of" instead of "each of".

-Line 146: Please delete the full stop "." after the word "degraded".

-Line 149: Please write at "incubation time" instead of "time" as well as "t [h]" instead of "t".

-Line 153: Please delete the word "by".

-Line 155: Delete the word "various".

-Lines 157-158: Please write "by inoculating (OD600 = 2.0)" instead of "by inoculating 2.0 (OD600)". Also, write the number 600 in subscript. 

-Line 163: Please write "inoculum concentrations" instead of "inoculums concentrations".

-Line 164: Write "inoculated with" instead of "inoculated".

-Line 165: Add space between "pH" and "7".

-Line 172: Write "H2SO4" instead of "sulphuric acid (H2SO4)" since you have given in Section 2.2 the full name of the formula.

-Line 184: What do you mean "as one"? You should be more specific. Can you give the initial concentration of the compound? Is there any available information in the literature regarding the compound concentration level that is responsible for the odor?

-Line 193: Can you explain why you selected only 6? Was it a random selection or based on differences in the morphological characteristics?

-Line 194: Delete the word "that".

-Line 200: Please add "spp." after each of the three bacterial species.

-Line 202: Please delete the word "population".

-Line 209: Measured where? On the pit latrine? Please be more specific.

-Line 211: The term "far less" is not specific. Can you write how many folds less?

-Line 212: Write "cultures are" instead of "cultures as".

-Lines 213, 214, 218: A Figure separated in parts (a) and (b) should have only one legend including both parts. Thus, add only one legend for Figure 2.

-Lines 227-228: These are the two bacteria with the highest degradation efficiency? Why did you choose to comment on only these two species?

-Line 228: Bacillus cereus should be written in italics.

-Lines 243-246: Can you mention the degradation efficiency of the same species in the work of Chin et al.? Were there any differences? Which compound concentration did he study? Also, had the above species similar degradation efficiency with the different species used in the other studies?

-Line 254: Write "are presented" instead of "as presented".

-Line 269: Write "is shown" instead of "as shown".

-Lines 277-278: Why did you suggest that the C3 was the best since C1 and C2 had also 100% degradation efficiency. It is not clear what you mean.

-Line 281: Please mention more clearly that you compare consortium C3 with the individual species. The word "individual" should be written.

-Lines 285, 293, 296: Again, add only one legend for Figure 4 including parts (a) and (b).

-Line 286: The rounded percentage is 99.9%, not 99.8%.

-Line 287: As I can see consortium C7 and C16 are almost the same considering the deviation of the measurements. Especially C7 (99.89%) is the same since the difference 0.11% is a part of the standard deviation. Please rewrite the paragraph accordingly.

-Line 323: How many folds better was the degradation level?

-Line 324: How many folds higher?

-Lines 350-351: According to my previous comments and Table 2, the consortiums C1, C2, C3 and C7 were the most effective. Write something else in the title. You could write "by effective bacterial strains" instead of "by bacterial strains of the most effective bacterial consortium". 

-Line 354: Again, rewrite the sentence since C3 was not the only effective consortium.

-Line 362: Write "various incubation" instead of "incubation".

-Lines 379, 416, 442: Explain what the abbreviations "SM" and "BC" stand for in the Figure legend.

-Line 412: There is a space in front of the word "their". Please correct it.

-Line 435: You have tested only one concentration above 2.0. Thus, don't write "any increase" but "an increase".

-Line 437: Please mention also that the optimum inoculum size for Serratia marcescens is 2.5.

-Line 438: Please delete the abbreviation "DMP". It's unnecessary since you are not using it in the rest of the manuscript.

-Line 457: Delete the word "as".

-Line 458: The word "totalling" is not a good use of the English language. Please improve the phrase.

-Line 459: One more time, rewrite the statement since consortiums C1, C2 and C7 had also the same effect. Do not use the word "the best" but write that it was "an effective".

-Line 464: The optimum temperature was "30 oC" not "35 oC". Please correct it.

-Lines 466-467: You wrote the opposite. According to Figure 7, size "2.0" was optimum for B. cereus and "2.5" was optimum for S. marcescens.

Author Response

Dear Sir or Madam,

Kind regards,

John
